# A Novel Anti-CD44 Variant 9 Monoclonal Antibody C$_{44}$Mab-1 Was Developed for Immunohistochemical Analyses against Colorectal Cancers

Mayuki Tawara [1,†], Hiroyuki Suzuki [1,*,†], Nohara Goto [1], Tomohiro Tanaka [1], Mika K. Kaneko [1,2] and Yukinari Kato [1,2,*]

1 Department of Molecular Pharmacology, Tohoku University Graduate School of Medicine, 2-1 Seiryo-machi, Aoba-ku, Sendai 980-8575, Japan; tawara.mayuki.p8@dc.tohoku.ac.jp (M.T.); s1930550@s.tsukuba.ac.jp (N.G.); tomohiro.tanaka.b5@tohoku.ac.jp (T.T.); k.mika@med.tohoku.ac.jp (M.K.K.)

2 Department of Antibody Drug Development, Tohoku University Graduate School of Medicine, 2-1 Seiryo-machi, Aoba-ku, Sendai 980-8575, Japan

* Correspondence: hiroyuki.suzuki.b4@tohoku.ac.jp (H.S.); yukinari.kato.e6@tohoku.ac.jp (Y.K.); Tel.: +81-22-717-8207 (H.S. & Y.K.)

† These authors contributed equally to this work.

**Abstract:** Cluster of differentiation 44 (CD44) is a type I transmembrane glycoprotein and has been shown to be a cell surface marker of cancer stem-like cells in various cancers. In particular, the splicing variants of CD44 (CD44v) are overexpressed in cancers and play critical roles in cancer stemness, invasiveness, and resistance to chemotherapy and radiotherapy. Therefore, the understanding of the function of each CD44v is indispensable for CD44-targeting therapy. CD44v9 contains the variant 9-encoded region, and its expression predicts poor prognosis in patients with various cancers. CD44v9 plays critical roles in the malignant progression of tumors. Therefore, CD44v9 is a promising target for cancer diagnosis and therapy. Here, we developed sensitive and specific monoclonal antibodies (mAbs) against CD44 by immunizing mice with CD44v3–10-overexpressed Chinese hamster ovary-K1 (CHO/CD44v3–10) cells. We first determined their critical epitopes using enzyme-linked immunosorbent assay and characterized their applications as flow cytometry, western blotting, and immunohistochemistry. One of the established clones, C$_{44}$Mab-1 (IgG$_1$, kappa), reacted with a peptide of the variant 9-encoded region, indicating that C$_{44}$Mab-1 recognizes CD44v9. C$_{44}$Mab-1 could recognize CHO/CD44v3–10 cells or colorectal cancer cell lines (COLO201 and COLO205) in flow cytometric analysis. The apparent dissociation constant ($K_D$) of C$_{44}$Mab-1 for CHO/CD44v3–10, COLO201, and COLO205 was $2.5 \times 10^{-8}$ M, $3.3 \times 10^{-8}$ M, and $6.5 \times 10^{-8}$ M, respectively. Furthermore, C$_{44}$Mab-1 was able to detect the CD44v3–10 in western blotting and the endogenous CD44v9 in immunohistochemistry using colorectal cancer tissues. These results indicated that C$_{44}$Mab-1 is useful for detecting CD44v9 not only in flow cytometry or western blotting but also in immunohistochemistry against colorectal cancers.

**Keywords:** CD44; CD44v9; monoclonal antibody; colorectal cancer



## 1. Introduction

Cluster of differentiation 44 (CD44) is a type I transmembrane glycoprotein, and its variety of isoforms are expressed in various types of cells [1]. The alternative splicing of CD44 mRNA mediates the variety of isoforms [2]. The CD44 standard (CD44s) isoform, the smallest isoform of CD44, is expressed in most vertebrate cells. CD44s mRNA is assembled by the first five (1 to 5) and the last five (16 to 20) constant region exons [3]. The CD44 variant (CD44v) isoforms are assembled by the alternative splicing of middle variant exons (v1–v10) in various combinations with the standard exons of CD44s [4]. Both CD44s and CD44v (pan-CD44) bind to hyaluronic acid (HA), which plays critical roles in cellular adhesion, migration, homing, and proliferation [5].

The CD44 protein is further modified using a variety of glycosyltransferases [6]. Due to the post-translational modifications, including *N*-glycans, *O*-glycans, and glycosaminoglycans (heparan sulphate, etc.), the molecular weight of CD44s is enlarged to 80–100 kDa, and some CD44v isoforms surpass 200 kDa due to a high level of glycosylation [7].

Several isoforms of the CD44 are associated with malignant progression in various tumors [8], including head and neck squamous cell carcinomas (HNSCCs) [9], pancreatic cancers [10,11], breast cancers [12], gliomas [13,14], prostate cancers [15], and colorectal cancers (CRC) [16]. CD44 is also known as a cell surface marker of cancer stem-like cells (CSCs) in various carcinomas [17]. Specific monoclonal antibodies (mAbs) to CD44s or CD44v are utilized for sorting CD44$^{high}$ CSCs [17]. The CD44$^{high}$ population exhibited the increased stemness property, drug resistance, and tumor formation in vivo [17]. Therefore, development of anti-CD44 mAbs, which recognize each variant, is important for the further characterization of CSCs in various cancers.

The functions of CD44v have been reported in the promotion of tumor invasion, metastasis, CSC properties [18], and resistance to chemotherapy and radiotherapy [8,19]. The v3-encoded region is modified by heparan sulfate, which promotes the binding to heparin-binding growth factors, including fibroblast growth factors and heparin-binding epidermal growth factor-like growth factor. Therefore, the v3-encoded region functions as a co-receptor of receptor tyrosine kinases and potentiates their signal transduction [20]. Furthermore, the v6-encoded region is essential for the activation of c-MET through ternary complex formation with the ligand hepatocyte growth factor [21]. The v8–10-encoded region could bind to and stabilize a cystine–glutamate transporter (xCT), which promotes the defense to reactive oxygen species (ROS) via cystine uptake-mediated glutathione synthesis [22]. The regulation of redox status depends on the expression of CD44v8–10 that is associated with the xCT function and links to the poor prognosis of patients [23]. Therefore, the establishment and characterization of mAbs, which recognize each CD44v, are essential for understanding each variant function and development of CD44-targeting cancer therapy. However, the function and distribution of the variant 9-encoded region in tumors have not been fully understood.

We previously developed an anti-pan-CD44 mAb, C$_{44}$Mab-5 (IgG$_1$, kappa) [24], using the Cell-Based Immunization and Screening (CBIS) method. Furthermore, another anti-pan-CD44 mAb, C$_{44}$Mab-46 (IgG$_1$, kappa) [25], was established by immunizing mice with CD44v3–10 ectodomain. We showed that both C$_{44}$Mab-5 and C$_{44}$Mab-46 could be applied to flow cytometry and immunohistochemistry in oral [24] and esophageal SCCs [25]. We also determined the epitopes of C$_{44}$Mab-5 and C$_{44}$Mab-46 within the standard exons (1 to 5)-encoding regions [26–28]. Furthermore, we produced a defucosylated version (5-mG$_{2a}$-f) using FUT8-deficient ExpiCHO-S cells (BINDS-09) and investigated the antitumor effects of 5-mG$_{2a}$-f in mouse xenograft models of oral SCC [29]. Recently, we have established various CD44v mAbs, including C$_{44}$Mab-108 (v4) [30], C$_{44}$Mab-3 (v5) [31], C$_{44}$Mab-9 (v6) [32], and C$_{44}$Mab-34 (v7/8) [33].

In this study, we established a novel anti-CD44v9 mAb, C$_{44}$Mab-1 (IgG$_1$, kappa), using the CBIS method and evaluated its applications for flow cytometry, western blotting, and immunohistochemical analyses of oral squamous cell carcinoma and colorectal adenocarcinomas.

## 2. Materials and Methods

### 2.1. Cell Lines

COLO201 (a human colorectal cancer cell line), P3X63Ag8U.1 (P3U1; a mouse multiple myeloma), and Chinese hamster ovary (CHO)-K1 cell lines were obtained from the American Type Culture Collection (ATCC, Manassas, VA, USA). COLO205 (a human colorectal cancer cell line) was obtained from the Cell Resource Center for Biomedical Research Institute of Development, Aging, and Cancer at Tohoku University (Miyagi, Japan). To cultivate these cell lines, we used Roswell Park Memorial Institute (RPMI)-1640 medium (Nacalai Tesque, Inc., Kyoto, Japan), which is supplemented with 10% heat-inactivated fetal

bovine serum (FBS; Thermo Fisher Scientific, Inc., Waltham, MA, USA). We further added the antibiotics, including 100 μg/mL streptomycin, 100 U/mL penicillin, and 0.25 μg/mL amphotericin B (Nacalai Tesque, Inc.). All cell lines were grown in a humidified incubator at 37°C with 5% $CO_2$.

We amplified CD44s cDNA from LN229 cDNA using the HotStar HiFidelity Polymerase Kit (Qiagen Inc., Hilden, Germany). We obtained CD44v3–10 ORF from the RIKEN BRC. CD44v3–10 and CD44s cDNAs were cloned into a pCAG-Ble-ssPA16 vector, which possesses the signal sequence and the N-terminal PA16 tag (GLEGGVAMPGAED-DVV) [24,34–37], which can be detected by an anti-human podoplanin (PDPN) mAb (NZ-1) [38–53]. Using a Neon transfection system (Thermo Fisher Scientific, Inc.), two stable transfectants, such as CHO/CD44v3–10 and CHO/CD44s, were established by introducing pCAG-Ble/PA16-CD44v3–10 and pCAG-Ble/PA16-CD44s into CHO-K1 cells, respectively.

### 2.2. Production of Hybridoma Cells

The 6-week-old female BALB/c mice were purchased from CLEA Japan (Tokyo, Japan). Mice were housed under specific pathogen-free conditions. To minimize animal suffering and distress in the laboratory, all mice experiments were performed according to relevant guidelines and regulations. Our animal experiments were approved by the Animal Care and Use Committee of Tohoku University (Permit number: 2019NiA-001). Mice were monitored every day for health during the period of experiments. Mice were intraperitoneally immunized with CHO/CD44v3–10 ($1 \times 10^8$ cells) with Imject Alum (Thermo Fisher Scientific Inc.) as an adjuvant. We performed additional immunizations of CHO/CD44v3–10 ($1 \times 10^8$ cells, three times) and performed a booster injection of CHO/CD44v3–10 ($1 \times 10^8$ cells) 2 days before harvesting the spleen cells. We used polyethylene glycol 1500 (PEG1500; Roche Diagnostics, Indianapolis, IN, USA) to fuse the splenocytes and P3U1 cells. The hybridoma supernatants, which are negative for CHO-K1 cells and positive for CHO/CD44v3–10 cells, were selected using SA3800 Cell Analyzer (Sony Corp. Tokyo, Japan).

### 2.3. ELISA

Fifty-eight peptides, which cover the extracellular domain of CD44v3–10 [26], were obtained from Sigma-Aldrich Corp. (St. Louis, MO, USA). We immobilized them on Nunc Maxisorp 96-well immunoplates (Thermo Fisher Scientific Inc) at 1 μg/mL for 30 min at 37 °C. The palate washing was performed using the HydroSpeed Microplate Washer (Tecan, Zürich, Switzerland) with phosphate-buffered saline (PBS) containing 0.05% (*v/v*) Tween 20 (PBST; Nacalai Tesque, Inc.). After the blocking with 1% (*w/v*) bovine serum albumin (BSA) in PBST for 30 min at 37 °C, $C_{44}$Mab-1 (10 μg/mL) was added to each well. Then, the wells were further incubated with anti-mouse immunoglobulins peroxidase-conjugate (1:2000 diluted; Agilent Technologies Inc., Santa Clara, CA, USA) for 30 min at 37 °C. One-Step Ultra TMB (Thermo Fisher Scientific Inc.) was used for enzymatic reactions. An iMark microplate reader (Bio-Rad Laboratories, Inc., Berkeley, CA, USA) was used to measure the optical density at 655 nm.

### 2.4. Flow Cytometry

CHO/CD44v3–10 and CHO-K1 cells were prepared using 0.25% trypsin and 1 mM ethylenediamine tetraacetic acid (EDTA; Nacalai Tesque, Inc.). COLO201 and COLO205 were obtained by pipetting. The cells ($1 \times 10^5$ cells/sample) were incubated with $C_{44}$Mab-1, $C_{44}$Mab-46, or blocking buffer (0.1% BSA in PBS; control) for 30 min at 4 °C. Then, the cells were treated with anti-mouse IgG conjugated with Alexa Fluor 488 (1:2000; Cell Signaling Technology, Inc.) for 30 min at 4 °C. Fluorescence data were collected and analyzed using the SA3800 Cell Analyzer and SA3800 software (ver. 2.05, Sony Corp.), respectively.

## 2.5. Determination of Apparent Dissociation Constant ($K_D$) by Flow Cytometry

In CHO/CD44v3–10 cells, we prepared from 260 to 0.016 nM (diluted by 1/2) of $C_{44}$Mab-1. In COLO201 and COLO205 cells, we prepared from 1300 to 0.08 nM (diluted by 1/2) of $C_{44}$Mab-1. The serially diluted $C_{44}$Mab-1 was suspended with $2 \times 10^5$ cells. Then, those cells were treated with anti-mouse IgG conjugated with Alexa Fluor 488 (1:200). Fluorescence data were collected and analyzed as indicated above. GraphPad Prism 8 (the fitting binding isotherms to built-in one-site binding models; GraphPad Software, Inc., La Jolla, CA, USA) was used to determine the apparent dissociation constant ($K_D$).

## 2.6. Western Blot Analysis

Cell lysates were prepared using NP-40 lysis buffer (20 mM tris-HCl [pH 7.5], 150 mM NaCl, 1% NP-40, and 50 μg/mL of aprotinin) and were boiled in sodium dodecyl sulfate (SDS) sample buffer (Nacalai Tesque, Inc.). The 10 μg of cell lysates were subjected to SDS-polyacrylamide gel for electrophoresis using polyacrylamide gels (5–20%; FUJIFILM Wako Pure Chemical Corporation, Osaka, Japan) and electrotransferred onto polyvinylidene difluoride (PVDF) membranes (Merck KGaA, Darmstadt, Germany). The blocking was performed using 4% skim milk (Nacalai Tesque, Inc.) in PBST. The membranes were incubated with 10 μg/mL of $C_{44}$Mab-1, 10 μg/mL of $C_{44}$Mab-46, or 1 μg/mL of an anti-isocitrate dehydrogenase 1 (IDH1; RcMab-1; rat IgG$_{2a}$) [54,55] and then incubated with peroxidase-conjugated anti-mouse immunoglobulins (diluted 1:1000; Agilent Technologies, Inc.) or peroxidase-conjugated anti-rat immunoglobulins (diluted 1:10,000; Sigma-Aldrich Corp.). Finally, the signals were enhanced using a chemiluminescence reagent, ImmunoStar LD (FUJIFILM Wako Pure Chemical Corporation), and detected using a Sayaca-Imager (DRC Co. Ltd., Tokyo, Japan).

## 2.7. Immunohistochemical Analysis

The formalin-fixed paraffin-embedded (FFPE) oral SCC tissues were obtained as described previously [56]. We purchased a colorectal carcinoma tissue array (CO483a) from US Biomax Inc. (Rockville, MD, USA). We used a cat rectum paraffin tissue section (Zyagen; FP-312) as a negative tissue control [57]. The sections were autoclaved in EnVision FLEX Target Retrieval Solution High pH (Agilent Technologies, Inc.) for 20 min. After blocking with SuperBlock T20 (Thermo Fisher Scientific, Inc.), we incubated the tissue sections with $C_{44}$Mab-1 (1 μg/mL) and $C_{44}$Mab-46 (1 μg/mL) for 1 h. For isotype control, we used PMab-44 (mouse IgG$_1$), an anti-bovine PDPN mAb [58]. The peptide blocking assay was performed as described previously [30]. The sections were further treated with the EnVision+ Kit for mouse (Agilent Technologies Inc.) for 30 min at room temperature. The chromogenic reaction was conducted using 3,3′-diaminobenzidine tetrahydrochloride (DAB; Agilent Technologies Inc.). The counterstaining was performed using hematoxylin (FUJIFILM Wako Pure Chemical Corporation). To examine the sections and obtain images, we used Leica DMD108 (Leica Microsystems GmbH, Wetzlar, Germany).

## 3. Results

### 3.1. Establishment of an Anti-CD44v9 mAb, $C_{44}$Mab-1

Figure 1A shows the structure of CD44s and representative CD44v. For the CBIS method, we prepared the CD44v3–10-overexpressed CHO-K1 cells (CHO/CD44v3–10) as an immunogen. Mice were immunized with CHO/CD44v3–10 cells (Figure 1B), and hybridomas were produced and seeded into 96-well plates (Figure 1C). Then, the supernatants, which were positive to CHO/CD44v3–10 cells and negative to CHO-K1, were selected by high throughput screening using flow cytometry (Figure 1D). After cloning by the limiting dilution, anti-CD44 mAb-producing clones were finally established (Figure 1E). We next performed the ELISA to determine the epitope of each mAb. Among them, $C_{44}$Mab-1 (IgG$_1$, kappa) was shown to recognize the CD44p471–490 peptide (STSHEGLEEDKDHPTTSTLT), which corresponds to the variant 9-encoded sequence (Supplementary Table S1). In con-

trast, $C_{44}$Mab-1 never recognized other CD44v3–10 extracellular regions. These results indicated that $C_{44}$Mab-1 specifically recognizes the CD44 variant 9-encoded sequence.

**Figure 1.** A schematic representation of anti-human CD44 mAbs production. (**A**) Structure of CD44. The CD44s mRNA is assembled by the first five (1 to 5) and the last five (16 to 20) exons and translates CD44s. The mRNAs of the CD44 variant are produced by the alternative splicing of middle variant exons and translate multiple CD44v such as CD44v3–10, CD44v4–10, CD44v6–10, and CD44v8–10. (**B**) CHO/CD44v3–10 cells were intraperitoneally injected into BALB/c mice. (**C**) Hybridomas were produced by fusion of the splenocytes and P3U1 cells. (**D**) The screening was performed by flow cytometry using CHO/CD44v3–10 and parental CHO-K1 cells. (**E**) After cloning and additional screening, a clone $C_{44}$Mab-1 (IgG$_1$, kappa) was established. Furthermore, we used peptides that cover the extracellular domain of CD44v3–10 (Supplementary Table S1) and determined the binding epitopes of each mAb using enzyme-linked immunosorbent assay (ELISA).

### 3.2. Flow Cytometric Analysis of C44Mab-1 to CD44-Expressing Cells

We next investigated the reactivity of C44Mab-1 against CHO/CD44v3–10 and CHO/CD44s cells using flow cytometry. C44Mab-1 recognized CHO/CD44v3–10 cells in a dose-dependent manner (Figure 2A). In contrast, C44Mab-1 never recognized CHO/CD44s (Figure 2B) or CHO-K1 (Figure 2C) cells. We confirmed that a pan-CD44 mAb, C44Mab-46 [25], recognized both CHO/CD44v3–10 and CHO/CD44s cells (Supplementary Figure S1A and B, respectively), but not CHO-K1 cells (Supplementary Figure S1C). Furthermore, C44Mab-1 could recognize endogenous CD44v9 in both COLO201 (Figure 2D) and COLO205 (Figure 2E) cells in a dose-dependent manner.

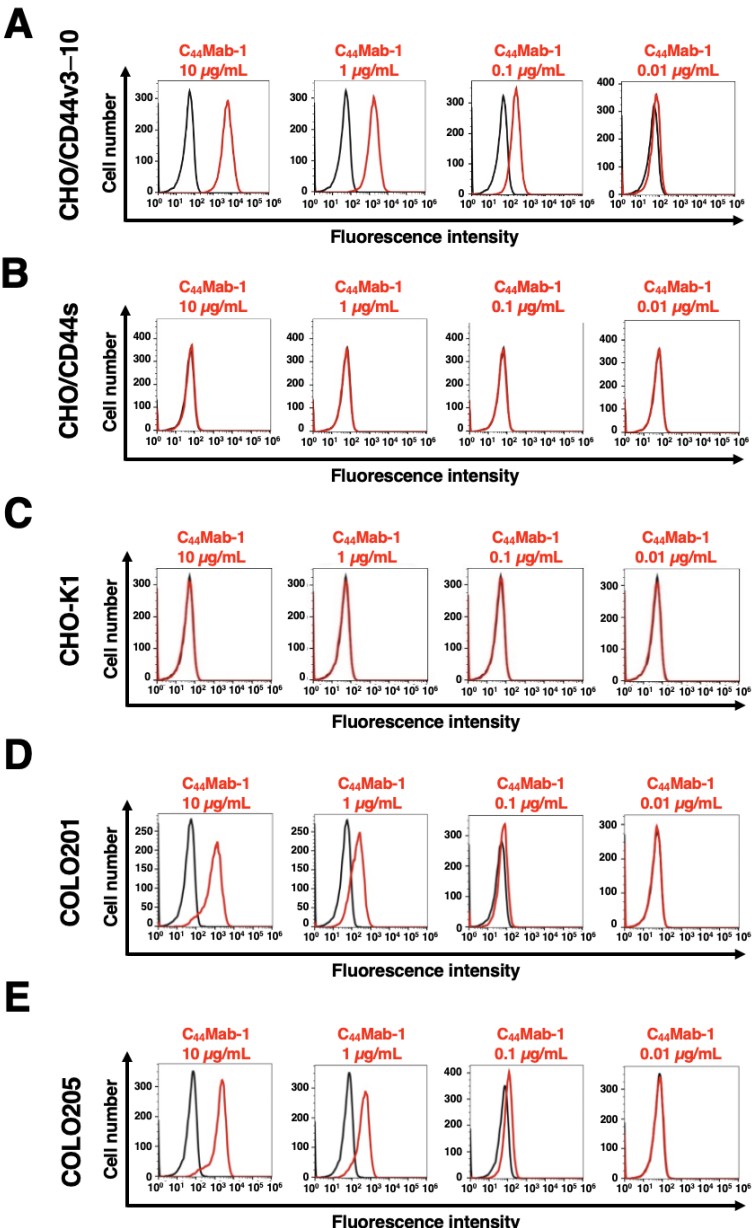

**Figure 2.** Flow cytometry using C44Mab-1. CHO/CD44v3–10 (**A**), CHO/CD44s (**B**), CHO-K1 (**C**), COLO201 (**D**), and COLO205 (**E**) were treated with 0.01–10 μg/mL of C44Mab-1, followed by treatment with Alexa Fluor 488-conjugated anti-mouse IgG (Red line). The black line represents the negative control (blocking buffer).

We next performed the flow cytometry-based measurement of the apparent binding affinity of C44Mab-1 to CHO/CD44v3–10, COLO201, and COLO205 cells. As shown in Figure 3, the dissociation constant ($K_D$) of C44Mab-1 for CHO/CD44v3–10 (Figure 3A),

COLO201 (Figure 3B), and COLO205 (Figure 3C) was $2.5 \times 10^{-8}$ M, $3.3 \times 10^{-8}$ M, and $6.5 \times 10^{-8}$ M, respectively. The results indicated that $C_{44}$Mab-1 possesses a moderate binding affinity for CD44v3–10 or endogenous CD44v9-expressing cells.

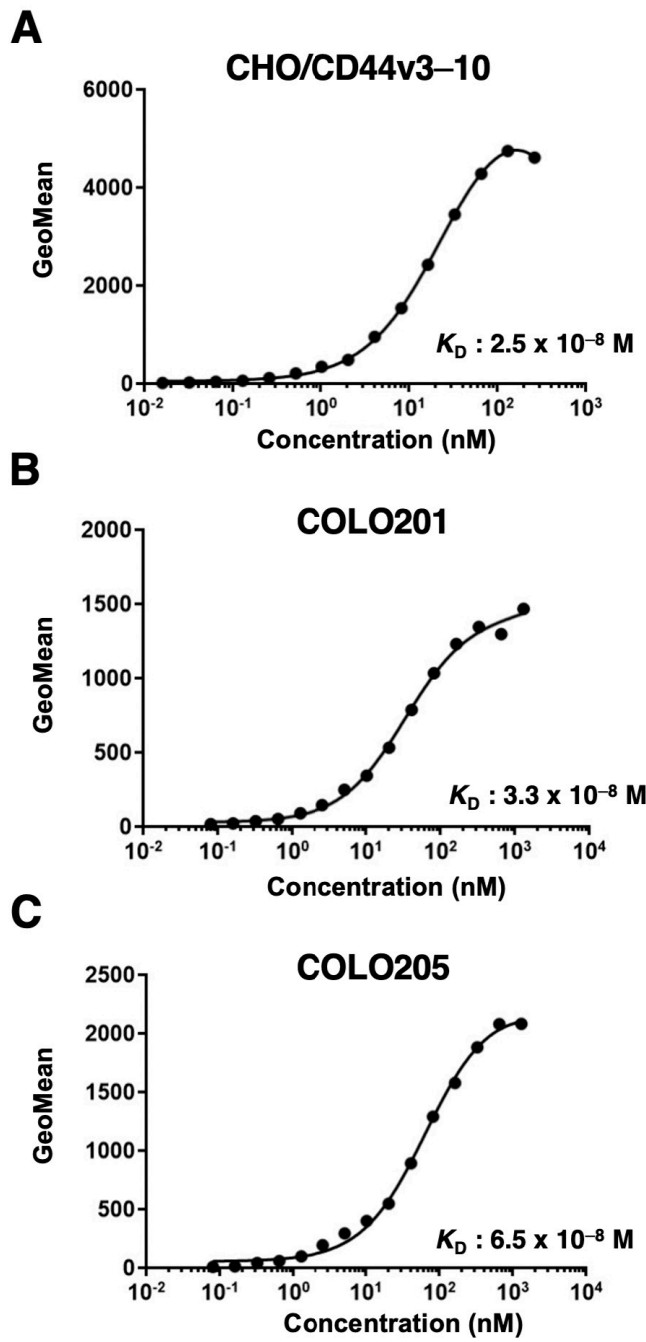

**Figure 3.** The determination of the binding affinity of $C_{44}$Mab-1. Serially diluted $C_{44}$Mab-1 at indicated concentrations was treated with CHO/CD44v3–10 (**A**), COLO201 (**B**), and COLO205 (**C**). Then, cells were treated with anti-mouse IgG conjugated with Alexa Fluor 488. Fluorescence data were collected, followed by the calculation of the apparent dissociation constant ($K_D$) by GraphPad PRISM 8.

*3.3. Western Blot Analysis*

We next performed western blot analysis to assess the sensitivity of $C_{44}$Mab-1. Total cell lysates of CHO-K1, CHO/CD44s, and CHO/CD44v3–10 were analyzed. As shown in Figure 4, $C_{44}$Mab-1 detected CD44v3–10 at more than 180-kDa and ~75 kDa bands mainly. However, $C_{44}$Mab-1 never detected any bands from lysates of CHO/CD44s and CHO-

K1 cells (Figure 4A). An anti-pan-CD44 mAb, $C_{44}$Mab-46, recognized CD44s (~75 kDa) and CD44v3–10 (>180 kDa) bands in the lysates of CHO/CD44s and CHO/CD44v3–10, respectively (Figure 4B). The loading control, IDH1 was observed in each lane (Figure 4C). These results indicated that $C_{44}$Mab-1 is able to detect exogenous CD44v3–10.

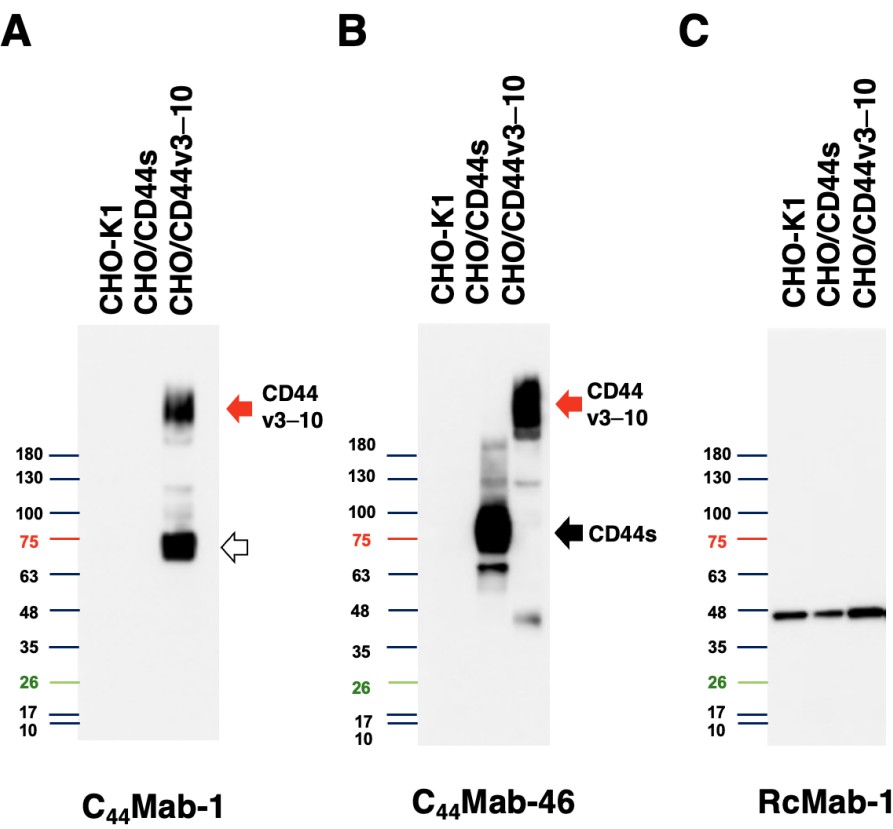

**Figure 4.** Western blot analysis by $C_{44}$Mab-1. The total cell lysates (10 μg of protein) were separated and transferred onto polyvinylidene difluoride (PVDF) membranes. The membranes were incubated with 10 μg/mL of $C_{44}$Mab-1 (**A**), 10 μg/mL of $C_{44}$Mab-46 (**B**), or 1 μg/mL of RcMab-1 (**C**), followed by incubation with peroxidase-conjugated anti-mouse (for $C_{44}$Mab-1 and $C_{44}$Mab-46) or anti-rat (for RcMab-1) immunoglobulins. The red arrows indicate the CD44v3–10 (>180 kDa). The black arrow indicates CD44s (~75 kDa). The white arrow indicates the lower molecular weight band recognized by $C_{44}$Mab-1 in CHO/CD44v3–10 lysate (~75 kDa).

### 3.4. Immunohistochemical Analysis Using $C_{44}$Mab-1 against Tumor Tissues

We next examined whether $C_{44}$Mab-1 could be used for immunohistochemical analyses using FFPE sections. Because HNSCC has been revealed as the second highest CD44-expressing cancer type in the Pan-Cancer Atlas [9], we first examined the reactivity of $C_{44}$Mab-1 and $C_{44}$Mab-46 in an oral SCC tissue, as a positive tissue control. As shown in Supplementary Figure S2, $C_{44}$Mab-1 exhibited a clear membranous staining and was able to clearly distinguish tumor cells from stromal tissues. In contrast, $C_{44}$Mab-46 stained both. The reactivity of $C_{44}$Mab-1 was completely blocked by the epitope peptide CD44p471–490 (Supplementary Figure S3A). Isotype control antibody (PMab-44, mouse IgG$_1$, kappa) did not stain the oral SCC tissue (Supplementary Figure S3B). $C_{44}$Mab-1 did not stain a negative tissue control (cat rectum, Supplementary Figure S3C).

We then investigated the reactivity of $C_{44}$Mab-1 and $C_{44}$Mab-46 in the CRC tissue array. $C_{44}$Mab-1 showed strong membranous and cytoplasmic staining throughout CRC cells (Figure 5A). $C_{44}$Mab-46 similarly stained the CRC cells (Figure 5B). In some CRC tissues, both $C_{44}$Mab-1 and $C_{44}$Mab-46 stained the basolateral surface of CRC cells (Figure 5C,D). In contrast, neither $C_{44}$Mab-1 nor $C_{44}$Mab-46 ever stained CRC cells in some CRC tissues

(Figure 5E,F). In addition, stromal staining by $C_{44}$Mab-46 was also observed in several tumor tissues (Figure 5F). In normal colon epithelium, epithelial cells were rarely stained by $C_{44}$Mab-1 (Figure 5G). In contrast, $C_{44}$Mab-46 mainly stained stromal tissues in normal colon epithelium (Figure 5H).

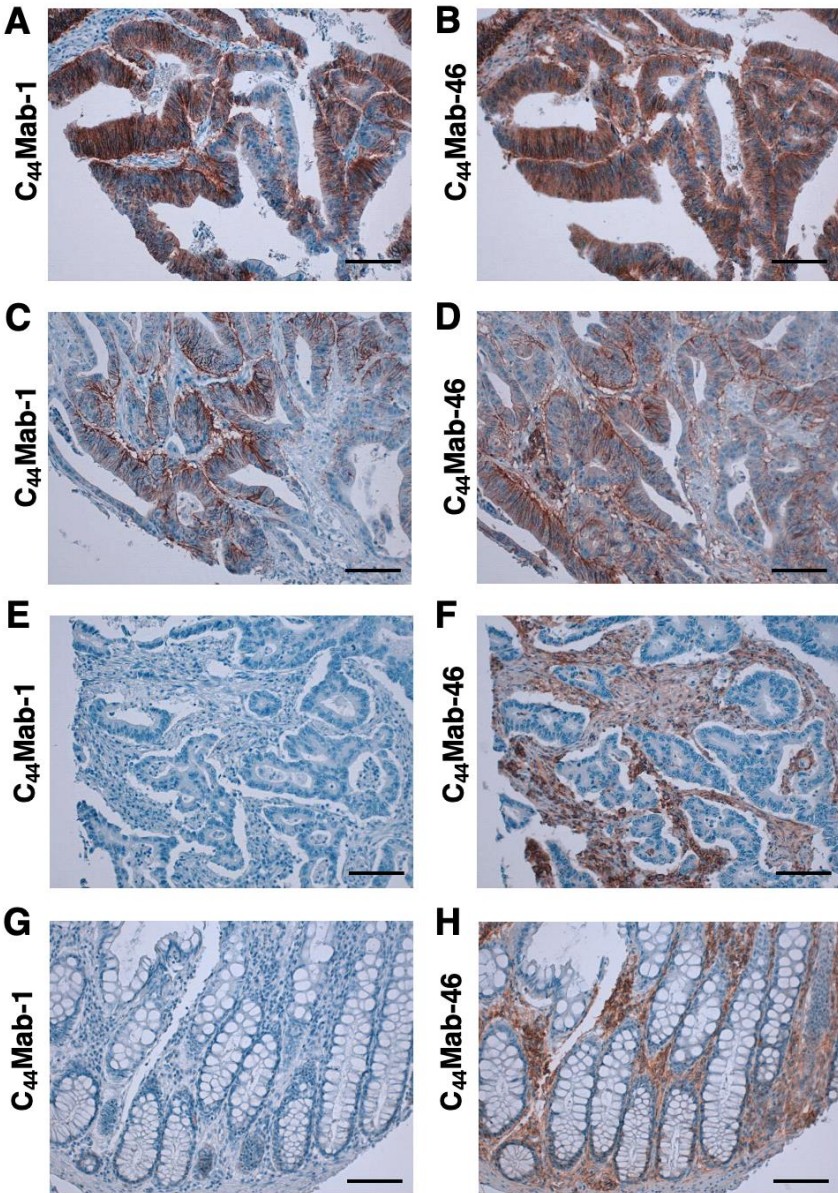

**Figure 5.** Immunohistochemical analysis using $C_{44}$Mab-1 and $C_{44}$Mab-46 against CRC tissues. After antigen retrieval, serial sections of CRC tissue arrays (CO483a) were incubated with 1 µg/mL of $C_{44}$Mab-1 or $C_{44}$Mab-46 followed by treatment with the Envision+ kit. The color was developed using 3,3'-diaminobenzidine tetrahydrochloride (DAB), and the sections were counterstained with hematoxylin. Scale bar = 100 µm. CRC (**A–F**); normal colon epithelium (**G,H**).

We summarized the data of the immunohistochemical analyses in Table 1; $C_{44}$Mab-1 stained 16 out of 40 cases (40%) in CRC. These results indicated that $C_{44}$Mab-1 is useful for immunohistochemical analysis of FFPE tumor sections.

**Table 1.** Immunohistochemical analysis using $C_{44}$Mab-1 against colorectal carcinoma tissue array.

| No. | Age | Sex | Organ | Pathology Diagnosis | Grade | Stage | Type | $C_{44}$Mab-1 | $C_{44}$Mab-46 |
|---|---|---|---|---|---|---|---|---|---|
| 1 | 67 | M | Colon | Adenocarcinoma | 1 | - | Malignant | + | + |
| 2 | 48 | M | Colon | Adenocarcinoma | 1 | IIA | Malignant | - | - |
| 3 | 58 | M | Colon | Adenocarcinoma | 1–2 | IIA | Malignant | + | + |
| 4 | 75 | M | Colon | Adenocarcinoma | 1 | IV | Malignant | - | ++ |
| 5 | 86 | M | Colon | Adenocarcinoma | 2 | II | Malignant | - | + |
| 6 | 55 | M | Colon | Adenocarcinoma | 2 | IIIC | Malignant | - | - |
| 7 | 38 | M | Colon | Adenocarcinoma | 1 | I | Malignant | - | ++ |
| 8 | 52 | M | Colon | Adenocarcinoma | 1 | IIIB | Malignant | + | - |
| 9 | 46 | M | Colon | Adenocarcinoma | 2 | IIIB | Malignant | ++ | + |
| 10 | 61 | M | Colon | Mucinous adenocarcinoma | 2 | IIIB | Malignant | + | ++ |
| 11 | 55 | M | Colon | Mucinous adenocarcinoma with necrosis | 2 | IIA | Malignant | - | ++ |
| 12 | 55 | M | Colon | Adenocarcinoma | 1 | IIIB | Malignant | + | - |
| 13 | 44 | M | Colon | Adenocarcinoma | 1 | - | Malignant | - | - |
| 14 | 31 | M | Colon | Adenocarcinoma | 2 | IIIB | Malignant | - | + |
| 15 | 74 | F | Colon | Adenocarcinoma | 2 | IIIB | Malignant | + | + |
| 16 | 61 | M | Colon | Adenocarcinoma | 2 | II | Malignant | ++ | ++ |
| 17 | 45 | M | Colon | Adenocarcinoma | 2 | III | Malignant | + | + |
| 18 | 58 | M | Colon | Adenocarcinoma | 2 | IIIB | Malignant | - | ++ |
| 19 | 58 | M | Colon | Adenocarcinoma | 2 | IIA | Malignant | +++ | +++ |
| 20 | 69 | M | Colon | Adenocarcinoma | 3 | - | Malignant | - | - |
| 21 | 64 | F | Colon | Adenocarcinoma | 2 | IIIC | Malignant | ++ | ++ |
| 22 | 82 | M | Colon | Adenocarcinoma | 2 | IIIB | Malignant | - | - |
| 23 | 34 | M | Colon | Adenocarcinoma | 2 | IIIB | Malignant | ++ | ++ |
| 24 | 50 | F | Colon | Adenocarcinoma | 2 | IIB | Malignant | - | - |
| 25 | 34 | F | Colon | Adenocarcinoma | 1 | IIB | Malignant | - | + |
| 26 | 52 | F | Colon | Adenocarcinoma | 2 | IIA | Malignant | - | + |
| 27 | 53 | F | Colon | Adenocarcinoma | 2 | IIIB | Malignant | - | - |
| 28 | 58 | F | Colon | Adenocarcinoma | 2 | I | Malignant | - | + |
| 29 | 59 | F | Colon | Adenocarcinoma | 2 | IIA | Malignant | ++ | ++ |
| 30 | 67 | M | Colon | Adenocarcinoma | 2 | IIIB | Malignant | - | ++ |
| 31 | 31 | M | Colon | Adenocarcinoma | 2 | IIIB | Malignant | +++ | +++ |
| 32 | 54 | F | Colon | Adenocarcinoma | 2 | IIB | Malignant | - | + |
| 33 | 54 | F | Colon | Adenocarcinoma | 2 | IIIB | Malignant | - | - |
| 34 | 62 | M | Colon | Adenocarcinoma | 2 | - | Malignant | - | + |
| 35 | 67 | F | Colon | Adenocarcinoma | 2 | - | Malignant | + | - |
| 36 | 52 | F | Colon | Adenocarcinoma | 2 | IIA | Malignant | - | - |
| 37 | 52 | F | Colon | Adenocarcinoma | 3 | IIIB | Malignant | - | - |
| 38 | 75 | M | Colon | Adenocarcinoma | 2 | - | Malignant | - | - |
| 39 | 57 | F | Colon | Adenocarcinoma | 2 | IIB | Malignant | + | +++ |
| 40 | 38 | M | Colon | Mucinous adenocarcinoma | 3 | I | Malignant | - | - |

-, No stain; +, Weak intensity; ++, Moderate intensity; +++, Strong intensity.

## 4. Discussion

Using the CBIS method, we developed C$_{44}$Mab-1 (Figure 1) and determined its epitope as a variant 9-encoded region using ELISA (Supplementary Table S1). Then, we showed the multiple applications of C$_{44}$Mab-1 for flow cytometry (Figures 2 and 3), western blotting (Figure 4), and immunohistochemistry using OSCC (Supplementary Figure S2) and CRC tissues (Figure 5 and Table 1).

Ishimoto et al. [22] demonstrated that CD44v interacts with xCT, a glutamate-cystine transporter, and regulates the level of reduced glutathione (GSH) in gastric cancer cells. As a result, CD44v contributes to the reduction of intracellular ROS. The knockdown of CD44 reduced the cell surface expression of xCT and suppressed tumor growth in a mouse gastric cancer model. Furthermore, they showed that the v8–10 region of CD44v is required for the specific interaction between CD44v and xCT, and CD44v8–10 (S301A), an *N*-linked glycosylation site mutant, failed to interact with xCT. These results showed an important function for CD44v in the regulation of ROS defense and tumor growth.

Ishimoto et al. [22] also established a rat mAb (clone RV3) against CD44v8–10 by immunizing CD44v8–10-expressed RH7777 cells. The epitope of the mAb was determined as a variant 9-encoded region using the recombinant CD44v9 protein by ELISA. RV3 was mainly used in immunohistochemistry and revealed a predictive marker for recurrence of gastric [59] and urothelial [60] cancers, predicting survival outcome in hepatocellular carcinomas [61], and an indicator for identifying a cisplatin-resistant population in urothelial cancers [62]. Therefore, CD44v9 is a critical biomarker to evaluate the malignancy and prognosis of tumors. Furthermore, sulfasalazine, an xCT inhibitor, was shown to suppress the survival of CD44v9-positive CSCs both in vitro [63–65] and in vivo [66]. A dose-escalation clinical study in patients with advanced gastric cancers revealed that sulfasalazine reduced the population of CD44v9-positive cells in tumors [67], suggesting that CD44v9 is a biomarker for patient selection and efficacy of xCT inhibitors.

As mentioned above, RV3 recognized the recombinant CD44v9 protein using ELISA. Therefore, RV3 is thought to recognize the peptide or glycopeptide structure of CD44v9. However, the detailed binding epitope of RV3 has not been determined. As shown in Supplementary Table S1, C$_{44}$Mab-1 recognized a synthetic peptide (CD44p471–490; STSHE-GLEEDKDHPTTSTLT) that possesses multiple predicted and confirmed *O*-glycan sites [68]. As shown in Figure 4A, C$_{44}$Mab-1 recognized a ~75kDa band in CHO/CD44v3–10 lysate, which is approximately identical to the predicted molecular weight of CD44v3–10 based on the amino acid length. Therefore, C$_{44}$Mab-1 could recognize CD44v3–10 regardless of the glycosylation. The detailed epitope mapping and the influence of the glycosylation on C$_{44}$Mab-1 recognition should be investigated in future studies.

Using large-scale genomic analyses, CRCs were classified into four subtypes: microsatellite instability immune, canonical, metabolic, and mesenchymal types [69]. Since the CD44v9 was upregulated in 40% of CRC tissues (Figure 5 and Table 1), the relationship to the subtypes should be determined. Additionally, the mechanism of CD44v9 upregulation, including the transcription and the v9 inclusion by alternative splicing, should be investigated. Wielenga et al. [70] demonstrated that CD44 is a target gene of Wnt/β-catenin in a mice intestinal tumor model, suggesting that β-catenin signaling pathway could upregulate CD44 transcription. However, the mechanism of the variant 9 inclusion during the CRC development remains to be determined.

In immunohistochemical analysis, we observed CD44v9 expression throughout CRC cells (Figure 5A) and on the basolateral surface of CRC cells (Figure 5C). The basolateral expression of CD44 was previously observed and shown to be co-localized with HA [71], EpCAM-Claudin-7 complex [72], and Annexin II [73]. Therefore, the basolateral expression of CD44 may function to promote HA/adhesion-mediated signal transduction and contribute CRC tumorigenesis.

Clinical trials of anti-pan CD44 and CD44v6 mAbs have been conducted [74]. RG7356, an anti-pan CD44 mAb, exhibited an acceptable safety profile. However, the trial was terminated because of no clinical and dose-response relationship with RG7356 [75]. Clinical trials

of an antibody-drug conjugate (ADC), an anti-CD44v6 mAb bivatuzumab−mertansine, were conducted. However, it failed due to the high toxicity to skin [76,77]. The anti-CD44v6 mAb is further developed to chimeric antigen receptor T (CAR-T) cell therapy. The CD44v6 CAR-T showed antitumor effects against primary human multiple myeloma and acute myeloid leukemia [78]. Furthermore, the CD44v6 CAR-T also suppressed the xenograft tumor growth of lung and ovarian carcinomas [79], which is expected for the application against solid tumors. Although CD44v9 is rarely detected in normal colon epithelium by $C_{44}$Mab-1, CD44v9 could be detected in other normal tissues, including oral squamous epithelium (Supplementary Figure S2). For the development of the therapeutic use of $C_{44}$Mab-1, further investigations are required to reduce the toxicity to the above tissues.

Because anti-CD44 mAbs could have side effects by affecting normal tissues, the clinical applications of anti-CD44 mAbs are still limited. We previously developed PDPN-targeting cancer-specific mAbs (CasMabs) [80–83] and podocalyxin-targeting CasMabs [84], which are currently being applied to CAR-T therapy in mice models [39,40,48]. These CasMabs recognize cancer-specific aberrant glycosylation of the target proteins [83]. It is worthwhile to establish cancer-specific anti-CD44 mAbs using the CasMab method. Anti-CD44 CasMabs production can be applicable as a basis for designing and optimizing potent immunotherapy modalities, including ADCs and CAR-T therapies.

## 5. Conclusions

An anti-CD44v9 mAb, $C_{44}$Mab-1 is useful for detecting CD44v9 in flow cytometry, western blotting, and immunohistochemistry.

**Supplementary Materials:** The following supporting information can be downloaded at: https://www.mdpi.com/article/10.3390/cimb45040238/s1. Figure S1: Conformation of the recognition of CHO/CD44s and CHO/CD44v3–10 by C44Mab-46 by flow cytometry. CHO/CD44v3–10 (A), CHO/CD44s (B), and CHO-K1 (C) were treated with 0.01-10 µg/mL of C44Mab-46, followed by treatment with Alexa Fluor 488-conjugated antimouse IgG (Red line). The black line represents the negative control (blocking buffer). Figure S2: Immunohistochemical analysis using C44Mab-1 and C44Mab-46 against oral squamous cell carcinoma tissues. After antigen retrieval, the sections were incubated with 1 µg/mL of C44Mab-1 (A) and 1 µg/mL of C44Mab-46 (B), followed by treatment with the Envision+ kit. The color was developed using 3,3′-diaminobenzidine tetrahydrochloride (DAB), and the sections were counterstained with hematoxylin. Scale bar = 100 µm. Figure S3: The blocking assay by an epitope peptide, isotype control, and negative tissue control in immunohistochemistry. (A) Blocking of the C44Mab-1 reactivity to oral SCC tissue (positive tissue control) by the CD44 peptide (aa 471–490) containing the C44Mab-1 epitope. After antigen retrieval, sections were incubated with C44Mab-1 (1 µg/mL) or C44Mab-1 (1 µg/mL) plus human CD44 peptide (aa 471–490, 10 µg/mL). (B) The oral SCC tissue section was incubated with an isotype control mAb, PMab-44 (1 µg/mL). (C) Anegative control tissue section (cat rectum) was incubated with C44Mab-1 (1 µg/mL). The tissues were further treated with the Envision+ kit. The color was developed using DAB, and sections were counterstained with hematoxylin. Scale bar = 100 µm. Table S1: The determination of the binding epitope of C44Mab-1 by ELISA.

**Author Contributions:** M.T., N.G. and T.T. performed the experiments. M.K.K. and Y.K. designed the experiments. M.T. and H.S. analyzed the data. M.T., H.S. and Y.K. wrote the manuscript. All authors have read and agreed to the published version of the manuscript.

**Funding:** This research was supported in part by Japan Agency for Medical Research and Development (AMED) under Grant Numbers JP22ama121008 (to Y.K.), JP22am0401013 (to Y.K.), JP22bm1004001 (to Y.K.), JP22ck0106730 (to Y.K.), and JP21am0101078 (to Y.K.) and by the Japan Society for the Promotion of Science (JSPS) Grants-in-Aid for Scientific Research (KAKENHI) under Grant Numbers 21K20789 (to T.T.), 22K06995 (to H.S.), 21K07168 (to M.K.K.), and 22K07224 (to Y.K.).

**Institutional Review Board Statement:** The animal study protocol was approved by the Animal Care and Use Committee of Tohoku University (Permit number: 2019NiA-001) for studies involving animals.

**Informed Consent Statement:** Not applicable.

**Data Availability Statement:** All related data and methods are presented in this paper. Additional inquiries should be addressed to the corresponding authors.

**Conflicts of Interest:** The authors declare no conflict of interest involving this article.

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
