# Peer review of "A Novel Anti-CD44 Variant 9 Monoclonal Antibody C44Mab-1 Was Developed for Immunohistochemical Analyses against Colorectal Cancers"

_cimb, doi:10.3390/cimb45040238_

Round 1
Reviewer 1 Report
The article is devoted to the development of a new antibody allowing the detection of variant 9 of CD44 protein in the tissue of colorectal carcinomas.
The authors have sufficient experience in this field. The work has been performed thoroughly and is described in sufficient detail.
However, I have got some comments:
I believe that the paper missed or did not present IHC controls that will help you determine the specificity of the observed antibody staining.
The results of Positive Tissue Controls, Negative Tissue Controls, Isotype Controls and Absorption Controls are required.
None
Author Response
The article is devoted to the development of a new antibody allowing the detection of variant 9 of CD44 protein in the tissue of colorectal carcinomas.
The authors have sufficient experience in this field. The work has been performed thoroughly and is described in sufficient detail.
However, I have got some comments:
I believe that the paper missed or did not present IHC controls that will help you determine the specificity of the observed antibody staining.
The results of Positive Tissue Controls, Negative Tissue Controls, Isotype Controls and Absorption Controls are required.
Because head and neck SCC is revealed as the second highest CD44-expressing cancer type in the Pan-Cancer Atlas (ref 9), we examined the reactivity of C44Mab-1 in an oral SCC tissue as a positive tissue control (Supplementary Figure S2).
We performed additional immunohistochemical analysis for Negative Tissue Controls, Isotype Controls and Absorption Controls (Supplementary Figure S3).
The reactivity of C44Mab-1 was completely blocked by the epitope peptide, CD44p471–490 (Supplementary Figure S3A). Isotype control antibody (PMab-44, mouse IgG1, kappa) did not stain the oral SCC tissue (Supplementary Figure S3B). C44Mab-1 did not stain a negative tissue control (cat rectum, Supplementary Figure S3C).
Reviewer 2 Report
In this manuscript, Tawara et al investigated the development of a new anti-CD44-V9 Mab for IHC analysis of Colorectal Cancers. The approach and methodology are clear and understandable. The work is quite interesting to the readers and will be valuable for new development in this area, therefore I would recommend this work for publication in CIMB with some minor corrections.
Line 27-28: “C44Mab-1 reacted with CHO/CD44v3–10 cells or colorectal cancer cell lines (COLO201 and COLO205) by flow cytometry.” The sentence is incomplete.
Line 139: How many CHO/CD44v3–10 and CHO-K1 cells were treated/obtained? mention the number.
Line 147: “Serially diluted C44Mab-1” Write the exact number of concentration and dilutions and how many CHO/CD44v3–10, COLO201, and COLO205 cells were treated.
Point 2.6, in western blot analysis, the sentence is starting with “The 10 μg of cell lysates were subjected….”, explain before what experiment was performed and from where this lysate is coming from. This will make more sense to the readership.
There should be a separate conclusion section after the discussion.
Author Response
In this manuscript, Tawara et al investigated the development of a new anti-CD44-V9 Mab for IHC analysis of Colorectal Cancers. The approach and methodology are clear and understandable. The work is quite interesting to the readers and will be valuable for new development in this area, therefore I would recommend this work for publication in CIMB with some minor corrections.
Line 27-28: “C44Mab-1 reacted with CHO/CD44v3–10 cells or colorectal cancer cell lines (COLO201 and COLO205) by flow cytometry.” The sentence is incomplete.
We changed the sentence.
Line 139: How many CHO/CD44v3–10 and CHO-K1 cells were treated/obtained? mention the number.
1 × 105 cells/sample. We added the information in section 2.4.
Line 147: “Serially diluted C44Mab-1” Write the exact number of concentration and dilutions and how many CHO/CD44v3–10, COLO201, and COLO205 cells were treated.
2 × 105 cells/sample.
In CHO/CD44v3–10 cells, we prepared from 260 to 0.016 nM (diluted by 1/2) of C44Mab-1.
In COLO201 and COLO205 cells, we prepared from 1300 to 0.08 nM (diluted by 1/2) of C44Mab-1.
We added the information in section 2.5.
Point 2.6, in western blot analysis, the sentence is starting with “The 10 μg of cell lysates were subjected….”, explain before what experiment was performed and from where this lysate is coming from. This will make more sense to the readership.
We added the information in section 2.6.
There should be a separate conclusion section after the discussion.
We added the conclusion section.
Round 2
Reviewer 1 Report
I have seen all the changes made by the author in response to my comments, so I believe the manuscript can be accepted for publication.